# RollingQ: Reviving the Cooperation Dynamics in Multimodal Transformer

**Haotian Ni** [1]   **Yake Wei** [2 3 4]   **Hang Liu** [5]   **Gong Chen** [6]   **Chong Peng** [6]   **Hao Lin** [6]
**Di Hu** [2 3 4]

## Abstract

Multimodal learning faces challenges in effectively fusing information from diverse modalities, especially when modality quality varies across samples. Dynamic fusion strategies, such as attention mechanism in Transformers, aim to address such challenges by adaptively emphasizing modalities based on the characteristics of input data. However, through amounts of carefully designed experiments, we surprisingly observed that the dynamic adaptability of widely-used self-attention models diminishes. Model tends to prefer one modality regardless of data characteristics. This bias triggers a self-reinforcing cycle that *progressively overemphasizes* the favored modality, widening the distribution gap in attention keys across modalities and deactivating attention mechanism's dynamic properties. To revive adaptability, we propose a simple yet effective method Rolling Query (RollingQ), which balances attention allocation by rotating the query to break the self-reinforcing cycle and mitigate the key distribution gap. Extensive experiments on various multimodal scenarios validate the effectiveness of RollingQ and the restoration of cooperation dynamics is pivotal for enhancing the broader capabilities of widely deployed multimodal Transformers. The source code is available at Github.

## 1. Introduction

Multimodal learning focuses on extracting and integrating information from data across different modalities (Bal-

---
[1]Beihang University, Beijing, China [2]Gaoling School of Artificial Intelligence Renmin University of China, Beiiing, China [3]Beijing Key Laboratory of Research on Large Models and Intelligent Governance, Beijing, China [4]Engineering Research Center of Next-Generation Intelligent Search and Recommendation. MOE, Beijing, China [5]Xiamen University, Xiamen, China [6]Tencent, Shenzhen, China. Correspondence to: Di Hu <dihu@ruc.edu.cn>.

*Proceedings of the $42^{nd}$ International Conference on Machine Learning*, Vancouver, Canada. PMLR 267, 2025. Copyright 2025 by the author(s).

trušaitis et al., 2018; Zhang et al., 2024) to achieve a comprehensive understanding. Each modality provides unique information, and the challenge of effectively fusing these modalities has been emphasized in prior works (Gao et al., 2020; Liang et al., 2022). Current research primarily explores two fusion paradigms: static fusion and dynamic fusion (Zhang et al., 2024). *Static fusion* applies fixed weights to different modalities during inference (illustrated in Figure 1(a)), assuming that the information provided by each modality remains consistent (Zhang et al., 2024). However, this assumption often breaks down in real-world scenarios, where modality quality can vary significantly across samples. *Dynamic fusion*, by contrast, directly addresses this issue by dynamically adjusting weights to each modality based on the specific characteristics of the input data (illustrated in Figure 1(b)). To enable dynamic fusion, multimodal Transformers (Vaswani, 2017; Xu et al., 2023) have emerged as a powerful scheme, leveraging the inherent attention mechanism to identify and focus on the most informative and task-relevant tokens in the input (Clark, 2019; Kovaleva, 2019). Using self-attention and cross-attention fusion layers, these models can adaptively and effectively integrate information from different modalities, achieving superior performance in tasks such as audio-visual learning (Nagrani et al., 2021; Chumachenko et al., 2022), sentiment analysis (Wankhade et al., 2022), and visual question answering (Yu et al., 2019).

To validate the practical performance of static and dynamic fusion paradigms, we conducted experiments on the widely used audio-visual dataset, Kinetic-Sound (Arandjelovic & Zisserman, 2017). Surprisingly, dynamic fusion implemented through a self-attention layer, often considered more flexible and adaptive, achieves an accuracy of *67.0*, which underperforms the static fusion's accuracy of *68.0*[1]. To investigate this unexpected outcome, we first examine whether this attention-based dynamic fusion paradigm can adaptively adjust weights as anticipated, by analyzing the attention scores in the model's fusion layer. However, as shown in Figure 1(c), the model allocates disproportionately high attention to the audio modality, regardless of data characteristics. To further verify this behavior, we replaced the audio input with Gaussian noise, which contains no information.

---
[1]See Table 1 for more details.

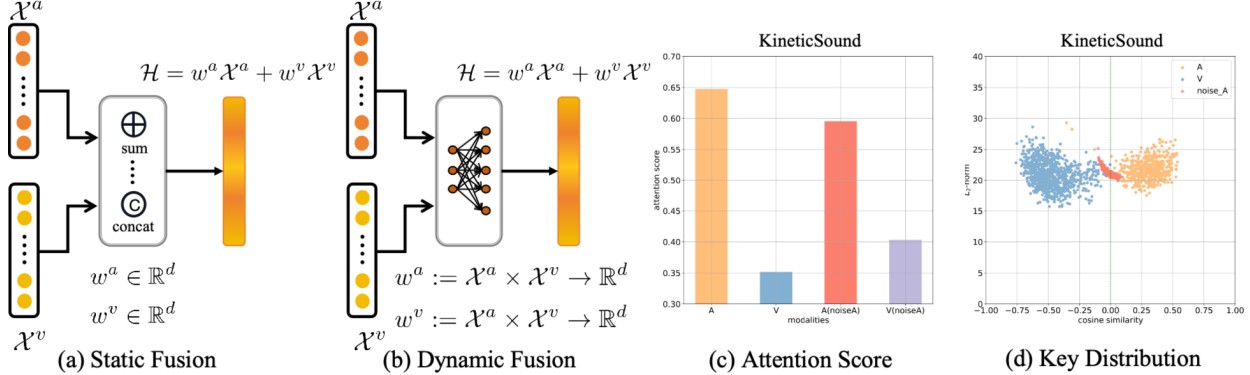

*Figure 1.* **(a)**: Illustration of multimodal static fusion, where the modality weights $\omega^a$ and $\omega^v$ are fixed values after learned. **(b)**: Illustration of multimodal dynamic fusion, where the modality weights $\omega^a$ and $\omega^v$ are determined by the input data $x^a$ and $x^v$. **(c)**: Attention scores assigned to each modality on the Kinetic-Sound dataset(Arandjelovic & Zisserman, 2017). The left half shows normal samples, while the right half shows samples where the audio modality's input is replaced with Gaussian noise. **(d)**: Distribution of keys in the attention layer for different modalities on Kinetic-Sound. The red dots represent the keys when the audio modality is replaced with Gaussian noise.

Despite this, the model remains excessive attention to the perturbed audio modality in Figure 1(c), indicating that such dynamic fusion paradigm is overly biased and fails to adapt attention score according to the informativeness of different modalities. This breakdown in multimodal attention adaptivity undermines the intended advantages of dynamic fusion and degrades overall model performance.

To investigate the underlying cause of this counterintuitive phenomenon, we analyzed the distribution of attention keys for each modality and their cosine similarity with the query of the class token, which directly determines the attention scores. As shown in Figure 1(d), the degradation of the dynamic property in the attention mechanism arises from a significant disparity in the attention key distributions across modalities. *A modality is biased*: the query of the class token, which determines the prediction of the model, remains significantly similar to the keys of the biased modality even when it contains no information. To further understand the factors driving this distribution gap, we performed both theoretical analysis and empirical verification. Our findings reveal *a self-reinforcing cycle* driven by the greedy nature of multimodal learning (Wu et al., 2022), *where the model continuously emphasizes the biased modality in feed-forward stage and optimizes its features through backward propagation.* Over time, this feedback loop exacerbates the distribution disparity: the biased modality accumulates higher attention scores and more informative features, while other modalities receive diminished attention and remain underoptimized. As a result, regardless of the data characteristics, the biased modality consistently receives higher attention scores due to the distribution gap, thereby significantly impairing the attention mechanism's dynamic adaptability.

To break the self-reinforcing cycle and revive the cooperative dynamics of multimodal Transformers, we propose a simple yet effective strategy: Rotating the Query! Con-

cretely, we could facilitate dynamic multimodal cooperation by mitigating the over-reliance on the biased modality. This **Rolling Q**uery (**RollingQ**) algorithm rotates the current query towards an anchor, which is designed to allocate higher attention scores to the unbiased modality rather than the biased one. After rotating the query to the anchor, the learning of this new query will encourage the key distribution across modalities to narrow the gap between them. Consequently, during optimization, the formerly underutilized modality gains more momentum, enabling it to obtain more informative features. This rebalancing gradually equalizes the quality of features across modalities. Furthermore, we validate the effectiveness of our RollingQ strategy through extensive experiments and provide detailed observations demonstrating how RollingQ restores the dynamic adaptability of multimodal Transformers.

Our contributions are summarized as follows: **Firstly**, we identify the deactivation of the dynamic property in multimodal Transformers, which we attribute to a self-reinforcing cycle during training, resulting in a significant distribution gap between the attention keys of different modalities. **Secondly**, we propose a simple but effective algorithm, RollingQ, which effectively disrupts this self-reinforcing cycle by rotating the query and rebalancing the attention mechanism. **Finally**, through extensive experiments, we demonstrate that RollingQ not only restores the cooperative dynamics of multimodal Transformers but also significantly improves their performance across diverse datasets.

## 2. Related Works

### 2.1. Multimodal Learning

Multimodal learning aims to leverage data from multiple modalities, where the fusion paradigm serves as the core to integrate multimodal information (Gao et al., 2020). Based

on the ideology of aggregate information within the feature space, several works explore various fusion schemes to effectively combine features under a late fusion structure (Zadeh et al., 2017; Liu et al., 2018). Besides, instead of simply aggregating, some works focus on enhancing the multimodal interaction thus fusing information within modalities at an earlier stage. Specifically, they have invented special transfer modules or exchange mechanisms between encoders to introduce interaction (Joze et al., 2020; Wang et al., 2020b). Meanwhile, behind these fusion structures, researchers found that there is always a modality favored by the model, hence it dominate the overall learning process, leading to imbalance multimodal learning (Wang et al., 2020a; Huang et al., 2022). To address this, several methods have been proposed to mitigate learning imbalance through optimization adjustments, data filtration, and specialized objectives (Peng et al., 2022; Fan et al., 2023; Xia et al., 2023; Wei et al., 2024; Yang et al., 2024; 2025; Huang et al., 2025). However, these approaches primarily focus on the static fusion paradigms. In this work, we examine a more complex paradigm, dynamic fusion, investigating the side effects of imbalanced feature quality on both attention score allocation and encoder optimization.

## 2.2. Multimodal Transformers

Multimodal Transformers (Xu et al., 2023), which leverage the attention mechanism to integrate information from multiple modalities, are widely used across various fields since they can dynamically select informative and task-relevant tokens to accomplish the task. Some works directly concatenate the input sequence of different modalities and pass them together into the Transformer blocks (Li et al., 2019; Kim et al., 2021). However, identifying the redundancy and computational expense associated with the long input sequence, others focus on structural modifications to unlock the full potential of Transformer models (Gao et al., 2019; Tsai et al., 2019; Wu et al., 2021). For example, some approaches use Transformer blocks to fuse features provided by two unimodal encoders (Chumachenko et al., 2022). Nagrani et al. (2021) and Recasens et al. (2023) limit modality interactions to specific tokens, thus compressing the interaction information for greater efficiency. Further, identifying that some tokens may be irrelevant to the task, Wang et al. (2022) propose a specialized scoring network to evaluate and select the most informative tokens for task completion.

Despite their reported superior performance on various tasks (Yu et al., 2019; Wankhade et al., 2022), the internal workings of the attention mechanism within fusion modules—and whether they function as intended—remain underexplored. In this work, we delve into the learning process of the attention module, shedding light on how the attention mechanism attributes weights across different modalities. We further propose the RollingQ algorithm, which restores

their dynamic capabilities and leads to better performance.

## 3. Method

### 3.1. Preliminary

In this work, we focus on exploring the attention mechanism within multimodal Transformers. To simplify the analysis and emphasize the core dynamics of attention, we use a single self-attention layer, which is both representative and widely adopted for theoretical analysis. As for different fusion paradigms within multi-layer Transformer, we also provide experiment verifications in Appendix C.4. For convenience, we denote the dataset as $D = \{x_i, y_i\}_{i \in [1,N]}$, where each sample in the dataset consists of a pair of different modalities $x_i = (x_i^a, x_i^v)$, using audio and visual data here for example. Two encoders are denoted by $\Phi^a, \Phi^v$, with parameter $\theta^a, \theta^v$, so the transformed features $z^a, z^v$ follows $z_i^m = \Phi^m(x_i^m; \theta^m), m \in \{a, v\}$. In order to apply the attention mechanism for fusion, the modality features should be a sequence of tokens with embedding dimension $d$ and sequence length $L^m$ for each modality, i.e. $z_i^m \in \mathbb{R}^{L^m \times d}, m \in \{a, v\}$. We follow Dosovitskiy (2020) and add a special $[class]$ token as a query vector denoted as $z_{cls} \in \mathbb{R}^d$, which will eventually be used as the output feature for task completion.

Within the attention layer (Vaswani, 2017), we have matrix $W^Q, W^K, W^V \in \mathbb{R}^{d \times d}$ that map current input into queries, keys, and values, respectively. We denote these as $K_i^m = z_i^m W^K, Q_i^m = z_i^m W^Q, V_i^m = z_i^m W^V, m \in \{a, v\}$. Specifically, we use $q = z_{cls} W^Q \in \mathbb{R}^d$ to denote the query for brevity. Hence, the output of the fusion layer can be simplified as follows:

$$A_i = \frac{q K_i^T}{\sqrt{d}}, \tag{1}$$

$$h_i = softmax(A_i) V_i, \tag{2}$$

where $K_i = [K_i^a, K_i^v], V_i = [V_i^a, V_i^v]$. Finally, we use a linear classifier $f(\cdot)$ to obtain the prediction $\hat{y}_i = f(h_i)$.

From Equation 1, we know that the attention score for each modality is determined by the dot product of the $q$ and $K_i^m$. Since $K_i^m \in \mathbb{R}^{L^m \times d}$ is a sequence of keys, we express it as $K_i^m = [k_{(i,1)}^m, ..., k_{(i,L^m)}^m]$. Similarly, the $A_i$ can be expanded into a sequence form as $A_i = [qk_{(i,1)}^a, ..., qk_{(i,L^a)}^a, qk_{(i,1)}^v, ..., qk_{(i,L^v)}^v]$.

Given that softmax function as a normalization function (Bridle, 1989), attention score for modality $m$ is determined by $\sum_{j=1}^{L^m} \frac{qk_{(i,j)}^m}{\sqrt{d}}$. By averaging the keys of modality $m$,

$$\hat{k}_i^m = \frac{\sum_{j=1}^{L^m} k_{(i,j)}^m}{L^m}, \quad m \in \{a, v\}, \tag{3}$$

we have:

$$\sum_{j=1}^{L^m} \frac{qk_{(i,j)}^m}{\sqrt{d}} = \frac{L^m}{\sqrt{d}}||q||_2||\hat{k}_i^m||_2 cos\theta_i^m. \quad (4)$$

Here, $||q||_2, ||\hat{k}_i^m||_2$ represent the $\mathcal{L}_2$-norms of $q, \hat{k}_i^m$, respectively, while $cos\theta_i^m$ denotes the cosine similarity between $q$ and $\hat{k}_i^m$.

### 3.2. A Self-reinforcing Cycle Undermines Cooperation Dynamics

To explore the underlying causes of the deactivated cooperation dynamics, we delve into the training process, providing a theoretical analysis of attention allocation during the feed-forward stage and the optimization of unimodal encoders during backward propagation.

*At the beginning of training*, since both modalities' features lack task-relevant information, the attention scores for each modality primarily depend on initialization. Specifically, we usually initialize the $[class]$ token using truncated normal distribution with expectation $\mathbb{E}[z_{cls}] = 0$ (Dosovitskiy, 2020). Inspired by Chen et al. (2018) and Geshkovski et al. (2023), we regard both query $q$ and the average key $\hat{k}_i^m$ as a sample from its corresponding random variables $Q, \hat{K}^m$ and distribution $\mathcal{Q}, \hat{\mathcal{K}}^m$. We can denote it as follows:

$$Q \sim \mathcal{Q}, \hat{K}^m \sim \hat{\mathcal{K}}^m. \quad (5)$$

From the $[class]$ token to query following $q = z_{cls}W^Q$, we has $\mathbb{E}[Q] = 0$. Due to the empirical fact that $Q$ and $\hat{K}^m$ are independent variables, their expectation can be directly multiplied, resulting in $\mathbb{E}[Q\hat{K}^a] = \mathbb{E}[Q\hat{K}^v] = 0$ at the training's initial phase. Therefore, we can deduce the following proposition:

**Proposition 3.1.** *At the start of training, both modalities receive similar attention scores.*

*As training progresses, the situation evolves during the feed-forward stage.* Due to intrinsic differences between modalities, a modality may be favored by the model and provide higher quality features over time, becoming the biased modality. Here, we assume audio is the biased modality.

Taking the greedy nature of multimodal deep neural networks into account (Wu et al., 2022), the model tends to prioritize the modality $a$ when it provides higher quality features. As a result, the biased modality accumulates higher attention scores, which are determined by $\sum_{j=1}^{L^a} \frac{qk_{(i,j)}^a}{\sqrt{d}}$. With Equation 4, this greedy nature increases cosine similarity between the modality $a$'s average key $\hat{k}_i^a$ and $q$ while simultaneously decreasing the cosine similarity between the modality $v$'s average key $\hat{k}_i^v$ and $q$. Consequently, this leads to a significant disparity in the average key distribu-

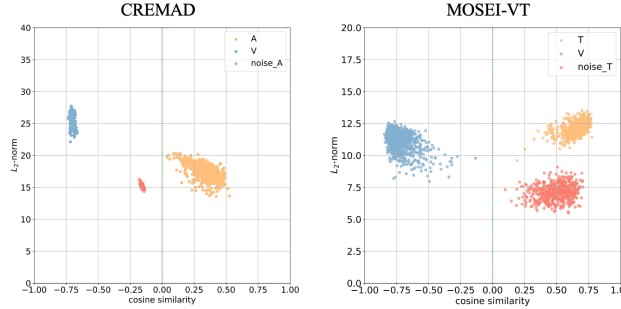

*Figure 2.* The distribution of average key on CREMA-D and MO-SEI(V+T); the x-axis refers to the cosine similarity with query while the y-axis refers to the $\mathcal{L}_2$-norm of key.

tions across modalities and disproportionately high attention score for biased modality.

*Meanwhile, modality bias will also influence the backward propagation for unimodal encoders.* We use $L(y; \hat{y})$ to denote the loss function where $y$ represents the ground truth label and $\hat{y}$ is the prediction generated by the linear classifier $f(\cdot)$, following $\hat{y} = f(h)$. We can integrate the Equation 1 and Equation 2 to the form below:

$$h_i = softmax(\frac{q[K_i^a, K_i^v]^T}{\sqrt{d}})[V_i^a, V_i^v], \quad (6)$$

the current loss is $L(y_i; f(h_i))$.

The gradient of modality $m$'s encoder parameter is:

$$\frac{\partial L}{\partial \theta^m} = \frac{\partial L}{\partial f} \frac{\partial f}{\partial h_i} \frac{\partial h_i}{\partial z_i^m} \frac{\partial z_i^m}{\partial \theta^m}. \quad (7)$$

According to Equation 7, we can find that $\frac{\partial L}{\partial f}$ and $\frac{\partial f}{\partial h_i}$ are shared for both modalities and $\frac{\partial z_i^m}{\partial \theta^m}$ depends on the modality-specific encoder $\Phi^m$ which is not influenced by the attention layer. Hence, the only difference in the gradients across modalities arises from $\frac{\partial h_i}{\partial z_i^m}$.

Using $s(\cdot)$ to denote $softmax(\cdot)$, with Equation 6, we can expand the gradient $\frac{\partial h_i}{\partial z_i^m}$ to

$$\frac{\partial s(\frac{q[K_i^a, K_i^v]^T}{\sqrt{d}})}{\partial K_i^m} \frac{\partial K_i^m}{\partial z_i^m} + s(\frac{q[K_i^a, K_i^v]^T}{\sqrt{d}}) \frac{\partial V_i^m}{\partial z_i^m}. \quad (8)$$

At the backward propagation for encoders, the biased modality $a$ receiving higher attention score $s(\frac{q[K_i^a, K_i^v]^T}{\sqrt{d}})$, has greater gradient $s(\frac{q[K_i^a, K_i^v]^T}{\sqrt{d}}) \frac{\partial V_i^a}{\partial z_i^a}$, augmenting the gradient of Equation 7 and Equation 8. It, in turn, reinforces the optimization of its corresponding unimodal encoder parameters $\theta^a$. In contrast, the unbiased modality $v$ receive less gradient from $s(\frac{q[K_i^a, K_i^v]^T}{\sqrt{d}}) \frac{\partial V_i^v}{\partial z_i^v}$, leading to under-optimized

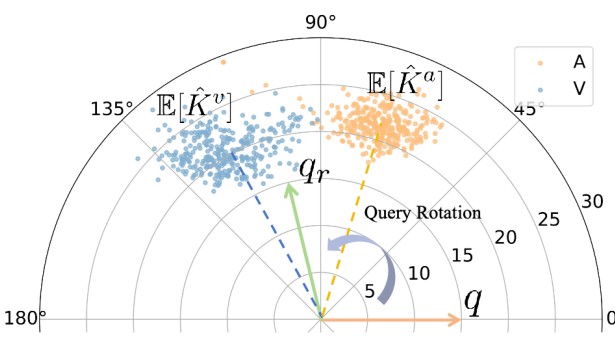

*Figure 3.* The illustration of RollingQ algorithm.

encoder parameters $\theta^v$. Consequently, this dynamic further exacerbates the inequality between feature qualities. To verify this analysis, we provide visualization on gradients of unimodal encoders in Appendix A.2.

*In summary, the modality bias triggers a self-reinforcing cycle:* during the feed-forward stage, biased modality accumulates more attention score due to its more informative feature, while in the backward propagation, the higher attention score provides more momentum to optimize the biased modality's encoder parameters. Consequently, this cycle not only creates a significant distribution disparity between $\hat{\mathcal{K}}^a$ and $\hat{\mathcal{K}}^v$ but also amplifies the inequality of feature quality between the two modalities.

Due to the significant distribution gap, the $\hat{k}_i^a$ and $\hat{k}_i^v$ maintain considerable differences regardless of data characteristics, where $\hat{k}_i^a$ consistently has higher cosine similarity with $q$ than $\hat{k}_i^a$. As shown in Figure 2, when the biased modality is replaced with Gaussian noise on the CREMA-D and CMU-MOSEI datasets, we observe a large distribution gap between the modalities. Notably, the noise input is mapped to a marginal position within the original distribution, yet it still exhibits a higher cosine similarity to $q$ compared to the unbiased modality. As a result, the model over-rely on the biased modality, leading to the deactivation of the cooperation dynamics in multimodal Transformers. Further verifications and visualizations on more datasets are provided in Appendix A.1.

### 3.3. Rolling Query Algorithm

Based on the above analysis, attention fusion tends to over-rely on the biased modality, losing its cooperation dynamics. It is therefore crucial to revive its dynamic property. To address this issue, we propose a simple yet effective **Rolling Query (RollingQ)** algorithm, which requires a few increase on model parameters and computation complexity[2].

The key idea to revive the cooperation dynamics is to find an effective method for identifying and breaking the self-

---

[2]See Table 4 for details

---

**Algorithm 1** Rolling Query Algorithm.

**Input:** Training Dataset $D = \{(x_i^a, x_i^v), y_i\}_{i=1,2,...,N}$, number of batch $N_B$, iter epoch $T$, hyperparameter $\rho, \beta$, model parameters $\theta^m, m \in \{a, v\}$, [class] token $z_{cls}$ and $W^Q, W^K$ in attention layer
Initialize $R = \mathbf{I}$
**for** $t = 1$ **to** $T$ **do**
    **for** $j = 1$ **to** $N_B$ **do**
        Sample Batch $B_j$
        Feed-forward with $q = z_{cls} W^Q R$
        Update model parameter $\theta^m$
        **if** $j = N_B$ **then**
            Calculate the average key using Equation 3
            Calculate $\mathbb{E}[\hat{K}^m] = mean(\hat{k}_i^m)$ in $B_j$
            Calculate $\mathbb{E}[Q] = mean(q)$ in $B_j$
            Calculate the AIR indicator by Equation 9
            **if** $|AIR| \geq \beta$ **then**
                Freeze the model parameter $\theta^m$
                Calculate $\alpha$ by Equation 11
                Calculate rebalance position $q_b$ by Equation 10
                Calculate rotation matrix $R_b$ by Equation 13
                Update rotation matrix $R = R(R_b.detach())$
                Unfreeze the model parameter $\theta^m$
            **end if**
        **end if**
    **end for**
**end for**

---

reinforcing cycle. Firstly, to detect the occurrence of this cycle, we can evaluate and monitor the distribution disparity across modalities. Next, to break the cycle, we must suppress its two driving factors: unequal feature quality and disproportionate attention scores. While feature quality is inherently determined by the modality itself, making it difficult to evaluate and control, we focus on modulating the attention scores. Specifically, we expect to find a reasonable anchor that could assign higher attention scores to the unbiased modality rather than the biased one. We then rotate the current query towards this anchor and let the new query learn in this region, encouraging the equalization of feature quality and reduction of distribution disparity across modalities.

**Evaluation of distribution gap.** Referring to Equation 3, we transform the attention score into a cosine similarity form. Based on the experiments observations and considering the LayerNorm layers used in the model, we hold $||\hat{k}_i^a||_2 \approx ||\hat{k}_i^v||_2, L^a \approx L^v$ in practice. Thus, the primary factor determining the attention score is the cosine similarity $cos\theta_i^m$ between the average key $\hat{k}_i^m$ of the modality $m$ and the query $q$. To quantify the distribution gap, we define the **A**ttention **I**mbalance **R**ate (**AIR**) indicator:

$$AIR = \mathbb{E}[cos\theta^a - cos\theta^v] \in [-2, 2]. \tag{9}$$

*Table 1.* Validation on CREMAD (Audio+Visual), Kinetic-Sound (Audio+Visual) and MOSEI (Visual+Text) with Transformer backbone. The best results are presented in bold. We use ↑ to show the improvement in performance compared to not implementing the RollingQ.

| | Dataset | Fusion | CREMA-D | Kinetic-Sound | CMU-MOSEI (V+T) |
| | Metric | Layers | Acc | Acc | Acc |
|---|---|---|---|---|---|
| Unimodal | Audio | 0 | 47.6 | 53.9 | - |
| | Visual | 0 | 36.3 | 57.0 | 47.1 |
| | Text | 0 | - | - | 60.9 |
| Static | Concat | 1 | 49.3 | 68.0 | 62.8 |
| | OGM (Peng et al., 2022) | 1 | 51.2 | 68.2 | 62.7 |
| | PMR (Fan et al., 2023) | 1 | 50.1 | 68.2 | 63.0 |
| Dynamic | Vanilla MT | 1 | 48.8 | 67.0 | 62.7 |
| | Vanilla MT* | 2 | 51.5 | 69.1 | 62.2 |
| | MulT (Tsai et al., 2019) | 1 | - | - | 62.4 |
| | MBT (Nagrani et al., 2021) | 2 | 51.5 | **72.2** | 63.0 |
| | JMT (Nagrani et al., 2021) | 2 | 50.7 | 67.7 | 62.6 |
| | MMML (Wu et al., 2024) | 2 | 52.0 | 69.8 | 62.8 |
| Ours | Vanilla MT+RollingQ | 1 | 51.9 (↑ 3.1) | 69.3 (↑ 2.3) | **63.2** (↑ 0.5) |
| | Vanilla MT*+RollingQ | 2 | 52.2 (↑ 0.7) | 70.1 (↑ 1.0) | 62.9 (↑ 0.7) |
| | MulT (Tsai et al., 2019)+RollingQ | 1 | - | - | 62.5 (↑ 0.1) |
| | MMML (Wu et al., 2024)+RollingQ | 2 | **52.7** (↑ 0.7) | 70.7 (↑ 0.9) | **63.2** (↑ 0.4) |

A hyperparameter $\beta$ serves as the threshold for $AIR$. For those $|AIR| \geq \beta$, we consider the distribution gap significant, indicating disproportionate attention scores.

**The balance anchor and the rotation matrix.** To break the self-reinforcing cycle, the query $q$ is rotated to a rebalanced anchor $q_b$ that could balance the allocation of attention scores and narrow the distribution gap between $\hat{\mathcal{K}}^a$ and $\hat{\mathcal{K}}^v$. A suitable position should favor the unbiased modality while considering the biased modality based on the AIR indicator. This enables the underutilized modality to gain more momentum during optimization, helping it learn higher quality features. As a result, the quality of features across modalities can gradually equalize during backward propagation, reducing the disparity in their key distributions during the feed-forward stage. To determine $q_b$, we calculate it using the expectation form, which allows for easy extension to multi-layer scenarios as we discussed in Appendix B, of the average key distributions $\mathbb{E}[\hat{K}^m]$ and the query distribution $\mathbb{E}[Q]$, with a weight $\alpha$:

$$q_b = (\alpha \frac{\mathbb{E}[\hat{K}^a]}{||\mathbb{E}[\hat{K}^a]||_2} + (1-\alpha) \frac{\mathbb{E}[\hat{K}^v]}{||\mathbb{E}[\hat{K}^v]||_2})||\mathbb{E}[Q]||_2, \quad (10)$$

while the weight $\alpha$ is derived, indicating $AIR$:

$$\alpha = \frac{1}{2}[1 + Tanh(-\rho AIR)], \quad (11)$$

where $\rho$ is an hyperparameter holds $\rho > 0$.

This ensures that the $q_b$ decreases the influence of the biased modality on the output. For instance, for $AIR > 0$ which indicates that modality $a$ is the biased modality, receiving a higher attention score than modality $v$, we will have

$Tanh(-\rho AIR) < 0$, resulting in $\alpha < 0.5$ and $1 - \alpha > 0.5$. Thus, the modality $a$ will have a lower impact on the determination of $q_b$ as shown in Equation 10.

To enable the current query to learn around the rebalance anchor as we wished, a rotation matrix $R_b \in \mathbb{R}^{d \times d}$ is essential. Since we already ensure the $\mathcal{L}_2$-norm of rebalance anchor $q_b$ and query $q$ is the same by Equation 10, we can calculate $R_b$ by Singular Value Decomposition(SVD) (Baker, 2005) with rebalance $q_b$ and the expectation of current query distribution $\mathbb{E}[Q]$:

$$R_b = SVD([\mathbb{E}[Q], q_b]), \quad (12)$$

which statisfied $q_b = \mathbb{E}[Q]R_b$.

Finally, for the query after rotation $q_r$, its value can be obtained by multiplying rotation matrix $R_b$ on $q$, where we can obtain:

$$q_r = qR_b. \quad (13)$$

Overall, by leveraging the AIR indicator to detect the self-reinforcing cycle and learning new rotated query around rebalance anchor with a rotation matrix, RollingQ can mitigate over-reliance on the biased modality and revive the cooperation dynamics in multimodal Transformer.

## 4. Experiments

### 4.1. Dataset and Experiment Settings

**Datasets. CREMA-D** (Cao et al., 2014) is an audio-visual dataset designed for emotion recognition, covering 6 common emotions. **Kinetic-Sound** (Arandjelovic & Zisserman,

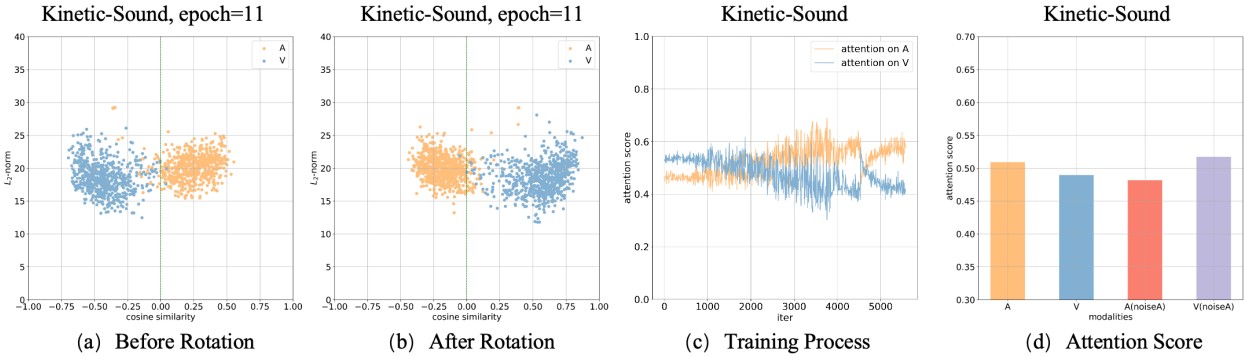

*Figure 4.* **(a)**: The distribution of two modalities before Rolling Query on the Kinetic-Sound dataset. **(b)**:The distribution of two modalities after Rolling Query on Kinetic-Sound dataset. **(c)**: The variation of attention score on each modality during the training process on Kinetic-Sound dataset when implementing the RollingQ algorithm. **(d)**: Attention scores assigned to each modality on the Kinetic-Sound dataset when implementing the RollingQ algorithm. The left half shows normal samples, while the right half shows samples where the audio modality's input is replaced with Gaussian noise.

2017) consists of 31 human action classes selected from the Kinetics dataset, which includes 400 action categories derived from YouTube videos. **CMU-MOSEI** (Zadeh et al., 2018) is a multimodal dataset that integrates audio, visual, and textual data.

**Settings** For the CREMA-D and Kinetic-Sound datasets, we use a 4-layer ViT-B/16 (Dosovitskiy, 2020) as the backbone, initializing it with pre-trained weights from ImageNet-21k (Ridnik et al., 2021). For CMU-MOSEI (Zadeh et al., 2018), we adopt a 4-layer vanilla Transformer following the preprocessing and settings outlined in Liang et al. (2021). As for training, we use SGD as optimizer and use cosine scheduler. The learning rate is set to 1e-3 with batch size 64 for all experiments. More detailed experimental settings can be found in Appendix C.1.

### 4.2. Comparison with Static and Dynamic Fusion

To validate the effectiveness of our RollingQ method in reviving the dynamic properties of the attention mechanism and improving overall performance, we conducted experiments on various datasets and compared our method with previous fusion techniques. We use vanilla concatenation as representations of static fusion approaches and include unimodal models as baselines. Besides, several methods, aiming at improving the performance of the static fusion paradigm by modulating the optimization for unimodal encoders, are considered, including OGM (Peng et al., 2022) and PMR (Fan et al., 2023). From the perspective of the dynamic fusion paradigm, we collect several attention-based methods from prior work for comparison, including MulT (Tsai et al., 2019), MBT (Nagrani et al., 2021), JMT (Waligora et al., 2024) and MMML (Wu et al., 2024). Finally, we use a single-layer attention for multimodal fusion method, named Vanilla MultiTrans, and a 2-layer Transformer fusion module, denoted by Vanilla Mul-

tiTrans*. We implement the RollingQ algorithm on these two representative methods to reveal its effectiveness and extend it to MulT (Tsai et al., 2019) and MMML (Wu et al., 2024) on its last attention layer to demonstrate the general applicability of our approach, with implementation details shown in Appendix B. The results are presented in Table 1.

Based on empirical results, we make several observations. First, compared to all unimodal models, multimodal models consistently outperform them, demonstrating that integrating information from multiple modalities is indeed beneficial. Besides, dynamic fusion methods do not always outperform static fusion methods, including the simple concatenation approach. This may be due to optimization challenges and the potential loss of dynamic properties as we explored in this work. Moreover, our RollingQ algorithm shows significant improvement over the original method, including the vanilla multimodal Transformers and specialized multimodal models like MulT (Tsai et al., 2019) and MMML (Wu et al., 2024), which indicates the prevalence of deactivated dynamic properties in multimodal Transformers and it is pivotal to revive the cooperation dynamics in multimodal Transformers. Finally, our method yields comparable results to imbalance techniques in static fusion scenarios, although RollingQ algorithm does not directly enhance the learning of unimodal encoders. Unlike more complex Transformer architectures, such as MBT (Nagrani et al., 2021) and JMT (Waligora et al., 2024), which require additional parameters and specialized modules, our RollingQ algorithm is grounded in a simple yet effective idea, demonstrating strong performance without the need for such complexities.

### 4.3. What Has RollingQ Done to Revive Cooperation Dynamics?

**Effective of rotation.** First, we visualize the distribution of average keys before and after the Rolling Query during

training. As shown in Figure 4(a) and Figure 4(b), after the Rolling Query, the biased audio modality and the unbiased visual modality exchange their similarity with the query. Specifically, the rotated query will learn around a new region, that could assign more attention to the visual modality, while slightly decreasing the similarity with the audio modality like Figure 4(b), which will not significantly impair the learning of audio modality but encourage the learning of visual modality. As a result, the audio modality, which still maintains more informative features, will no longer reinforce the optimization of its encoder, while the visual modality's optimization is enhanced.

**Evolution after rotation.** After rotating the query, the new query learns in a new rebalance region. Due to the fact that audio modality remains higher quality features, the new query tends to consecutively increase attention score to audio modality as we discussed in Section 3.2. To verify this analysis, we visualize the variation of attention scores on different modalities during the training process. As shown in Figure 4(c), the attention score of audio modality drops in certain iterations, while increasing latter. This phenomenon ensures that the query remains learnable based on the input data, indicating that the learning of attention allocation is not hindered by Rolling Query.

**Cooperation dynamics.** we investigate whether RollingQ can restore the cooperative dynamics of the attention mechanism. To test this, we replace the input of the biased modality with Gaussian noise and observe the results. First, by averaging the attention scores for both normal and noise samples, we can assess whether the attention layer assigns appropriate weights based on the current input. As shown in Figure 4(d), on the Kinetic-Sound dataset, replacing the audio modality with noise significantly reduces the attention score for the audio modality and increases the attention score for the visual modality. Compared to the disproportionately high attention observed in Figure 1(c), the model with the RollingQ algorithm demonstrates improved inter-modality cooperation. Further tests verifying cooperation dynamics based on attention like adding attention mask or QUAG

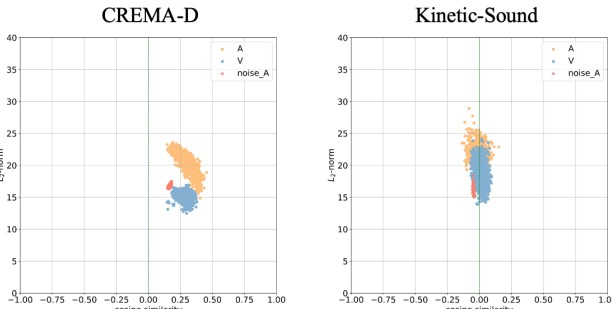

*Figure 5.* The distribution of two modalities is key when implementing the RollingQ algorithm. The red dots denote the audio modality with noise interference.

*Table 2.* The Pearson correlation between the attention score and whether it is a noise input. The coef closer to 1 or -1 indicates a stronger relation between attention score and input, while the $p < 0.01$ means that the coef is trustworthy.

| Dataset | CREMA-D | | Kinetic-Sound | |
|---|---|---|---|---|
| Correlation | coef | p | coef | p |
| Vanilla MT | 0.52 | $< 0.01$ | 0.44 | $< 0.01$ |
| Vanilla MT+RollingQ | 0.76 | $< 0.01$ | 0.78 | $< 0.01$ |

*Table 3.* Experiment results when perturbing the audio modality (the biased modality) with Gaussian noise following Liang et al. (2021) on Kinetic-Sound Dataset.

| Noise level | Vanilla MT | Vanilla MT + RollingQ |
|---|---|---|
| 0.00 | 67.0 | 69.3 |
| 0.25 | 62.7 ($\downarrow$ 4.3) | 67.2 ($\downarrow$ 1.9) |
| 0.50 | 52.9 ($\downarrow$ 14.1) | 58.2 ($\downarrow$ 11.1) |
| 0.75 | 43.2 ($\downarrow$ 23.8) | 47.5 ($\downarrow$ 21.8) |
| 1.00 | 34.7 ($\downarrow$ 32.3) | 40.6 ($\downarrow$ 28.7) |

attention (Rawal et al., 2023) are provided in Appendix C.2.

Additionally, we visualize the distribution of averaged attention keys across modalities. As shown in Figure 5, the RollingQ algorithm significantly reduces the distribution gap between modalities, compared to Figure 1(d). Regarding the noise input (represented by the red dots), it now shows a lower cosine similarity to the query than to the average keys of both modalities. Consequently, the attention mechanism allocates less attention to the relatively uninformative input in the biased modality as expected.

Furthermore, we conduct a correlation analysis between the noise input and attention scores in models with and without the RollingQ algorithm. Using Pearson correlation analysis on both the CREMA-D and Kinetic-Sound datasets, we assess whether the model can effectively recognize sample-wise variations in modality quality and adjust the attention scores accordingly. As shown in Table 2, after applying the RollingQ algorithm, the correlation coefficient increases significantly. This indicates that the attention mechanism becomes more sensitive to the data characteristics of each modality and can dynamically adjust attention scores. These findings confirm that the RollingQ algorithm successfully restores the cooperation dynamics in multimodal Transformer.

### 4.4. Test-time Adaptation for Noisy Biased Modality

To further and more comprehensively verify that RollingQ can restore cooperative dynamics within multimodal Transformers, thereby enhancing multimodal performance under more general conditions, we conduct experiments with noisy biased modality on Kinetic-Sound. We perturb the biased audio modality with Gaussian noise at various noise levels, as described in Liang et al. (2021), rendering the biased

Table 4. The accuracy, model parameters, and time complexity analysis on CREMA-D dataset. The GFLOPs are obtained from the thop library.

| Method | Acc | Parameters | GFLOPs |
|---|---|---|---|
| Vanilla MT | 48.8 | 59.87M | 1489.13 |
| MBT | 51.5 | 114.21M | 2746.90 |
| MMML | **52.0** | 77.88M | 1828.29 |
| JMT | 50.7 | 62.23M | 1494.87 |
| Vanilla MT + RollingQ | 51.9 | **60.46M** | **1489.20** |

Table 5. Abaltion of different batch size on CREMA-D dataset.

| Batch Size | Vanilla MT | Vanilla MT + RollingQ |
|---|---|---|
| 16 | 49.1 | 50.6 |
| 64 | 48.8 | 51.9 |
| 256 | 47.8 | 48.5 |

modality unreliable for final predictions. As shown in Table 3, both methods experience a performance drop as the noise level increases. However, RollingQ consistently outperforms the baseline, demonstrating its ability to better leverage both modalities and facilitate their cooperation. Besides, RollingQ exhibits a lower accuracy drop rate than the baseline, indicating that by reviving cooperation dynamics, RollingQ enables the model to better handle variations in data quality across modalities, enhancing its robustness to perturbations and leading to superior performance. Additionally, to evaluate RollingQ's performance under more severe and systematic distribution shifts, we conduct multimodal out-of-distribution (OOD) detection experiments following Dong et al. (2024) in Appendix C.3.

### 4.5. Complexity and Efficiency

Since RollingQ only needs an additional rotation matrix to work effectively, it is a simple method that requires few increases on parameters and computational cost. To verify the detailed additional cost of RollingQ, we record the model parameters and GFLOPs on experiments of CREMA-D dataset. As shown in Table 4, the proposed RollingQ algorithm requires *1% increase in parameters* and *0.1% increase on GFLOPs*, while achieving considerable improvements on its baseline vanilla multimodal Transformer and gaining comparable and even better performance over other special designed Transformer architectures.

### 4.6. Ablation Study

**Ablation of batch size**. To test the sensitivity of RollingQ toward batch size decision, we conducted an ablation study

Table 6. Ablation of different ViT encoder layers on Kinetic-Sound dataset.

| Encoder-Layer | Vanilla MT | Vanilla MT + RollingQ |
|---|---|---|
| 2 | 57.6 | 58.5 |
| 4 | 67.0 | 69.3 |
| 6 | 73.6 | 74.6 |

Table 7. Experiment results of our method with ResNet18 backbone on CREMA-D.

| Method | Acc |
|---|---|
| Audio | 60.1 |
| Visual | 40.1 |
| Concat | 61.4 |
| Vanilla MT | 60.8 |
| Vanilla MT + RollingQ | **62.6** |

on CREMA-D dataset with batch size around 16, 64, and 256 with all other hyperparameters fixed. As shown in Table 5, the proposed RollingQ algorithm consistently outperforms the baseline Vanilla MT by 0.7% - 3.1%. The results ensure the stability of RollingQ.

**Ablation of encoder layers**. To ensure that the analysis and effectiveness of RollingQ are consistent across different backbone scales, we conducted an ablation study on Kinetic-Sound dataset where the performance varied significantly across different encoder layers. As shown in Table 6, with encoder layers ranging from 2 4, and 6 and all other hyperparameters fixed, RollingQ consistently shows improvement compared to baseline Vanilla MT, verifying its effectiveness.

### 4.7. Adaptation to Other Backbone

To verify the generalization of our approach, we employed ResNet18 (He et al., 2016) and transformed the feature maps into tokens by flattening the temporal and spatial dimensions following (Carion et al., 2020; Huang et al., 2020). As shown in Table 7, our method consistently outperforms the baseline, demonstrating its effectiveness across different backbones and its potential to extend to various models.

## 5. Conclusion and Discussion

In this work, we identify the deactivation of cooperation dynamics in multimodal Transformer based on the observation of self-attention fusion layer, perform theoretical analysis and experimental verifications, and propose the RollingQ algorithm to restore its dynamic properties.

**Future work and limitations.** Our theoretical analysis focuses primarily on a single attention layer, while real-world Transformer models typically involve multiple layers, where the dynamics are more complex. Thus, further research is expected to explore how to better model, analyze, and restore cooperative dynamics in multi-layer Transformers. Additionally, our algorithm does not directly enhance the feature quality of unimodal encoders, which has been effectively achieved in previous static fusion approaches. Therefore, combining our method with those previous approaches to enhance and balance feature quality while addressing the deactivation issue is an avenue for future exploration.

## Acknowledgements

This work is supported by National Natural Science Foundation of China (NO.62106272). This work is also supported by Public Computing Cloud, Renmin University of China, and fund for building world-class universities (disciplines) of Renmin University of China.

## Impact Statement

This paper presents work whose goal is to advance the field of Machine Learning. There are many potential societal consequences of our work, none which we feel must be specifically highlighted here.

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

# A. Assumption and Analysis

## A.1. Attention and Average Key Distribution

To holistically and systematically validate our assumption and analysis discussed in Section 3.2, which mainly states that the modality bias in the multimodal training process triggers a self-reinforcing cycle that leads to inequality in feature quality and unreasonable attention score, we conduct experiments on more datasets including CREMA-D, Kinetic-Sound, CMU-MOSEI(A+T), CMU-MOSEI(V+T), UCF-101(Soomro et al., 2012), HMDB51(Kuehne et al., 2011), whose modality ranging from audio, RGB, text and optical flows and exhibiting variation in modality sequence length.

As shown in Figure 6, with different combinations of modalities and data set scales, the attribution of the unreasonable attention score agrees with our previous study. Besides, the visualization of average key distribution, where the noise input consistently has higher cosine similarity, further validates our assumption and theoretic analysis towards cooperation dynamics under imbalance multimodal learning.

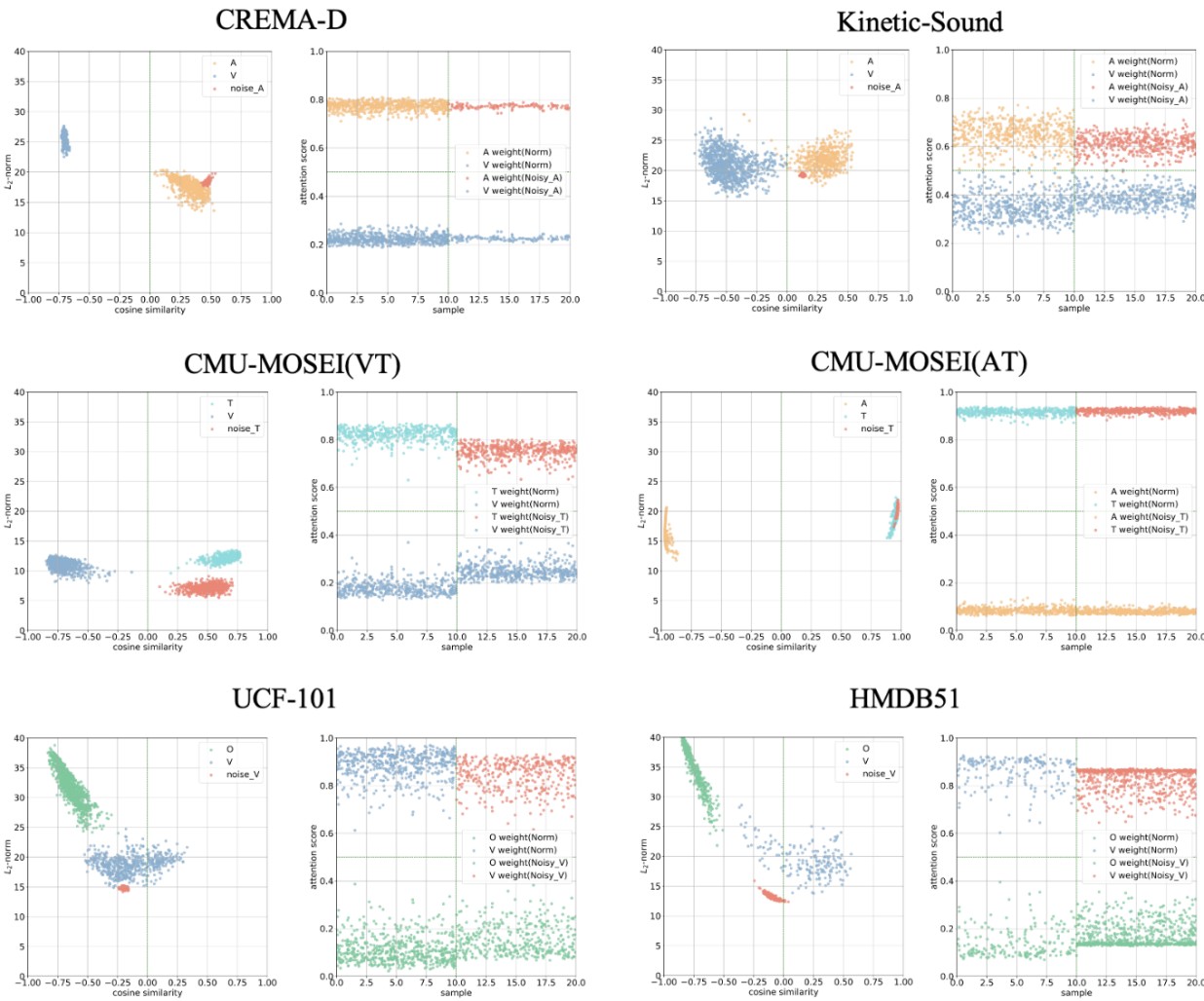

*Figure 6.* The visualization of attention score and average key distribution on CREMA-D, Kinetic-Sound, CMU-MOSEI(V+T), CMU-MOSEI(A+T), UCF-101, HMDB51 datasets, including audio (A), RGB (V), text (T) and optical flow (O) modalities. The left figure for each dataset denotes the average key distribution of different modalities and the red dots represent samples replacing the biased modality by Gaussian noise. The right figure for each dataset reveals the attention score for each modalities with normal or noise inputs.

## A.2. Gradients Evolution of Unimodal Encoders

To verify the analysis on gradients posted in Section 3.2, we conduct experiments on CREMA-D and Kinetic-Sound to monitor the gradients of each unimodal encoder. As shown in Figure 7, the gradient of the audio encoder increases significantly during the mid-stage, when the attention score begins to accumulate in the biased modality. This results in a noticeable gap between the modalities. As the total loss decreases over time, the gradients of both modalities drop, but their relative relationship remains. This further validates our theoretical analysis, showing that the biased modality consistently receives more optimization momentum. The gradient of the weaker modality never exceeds that of the biased modality, which can be explained by Equations 7 and 8. Since the only difference in gradients for each modality is $\frac{\partial h_i}{\partial z_i^m}$ and the loss is the multimodal loss, where "one modality becoming converged" equals "the multimodal model becoming converged," the total loss becomes very small. This small loss cannot provide enough momentum for the biased modality to optimize effectively. Hence, the gradient of the biased modality will consistently greater than the weak one.

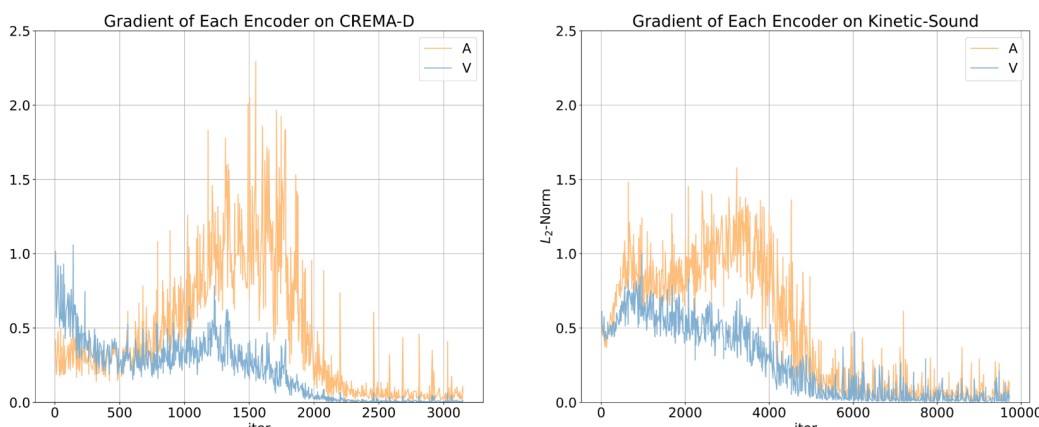

Figure 7. The $L_2$-Norm of gradient of the audio encoder and the visual encoder on CREMA-D and Kinetic-Sound datasets.

# B. Implement Details of The RollingQ Algorithm

**Rotation time limits for training stability.** In Algorithm 1, we calculate and update the rotation matrix whenever the absolute value of the AIR indicator exceeds its threshold. However, as $\beta$ becomes smaller, and the time required for rotation increases significantly. Based on our observation, for all dataset the RollingQ algorithm will rotate the query for about 3-5 times. However, the frequently changing query vector will impair training stability, resulting an not optimum performance. To address this, we set a maximum rotation limit in the RollingQ algorithm to ensure more stable learning. Specifically, the maximum rotation time is set to 1 for CREMA-D and MOSEI, and to 3 for Kinetic-Sound, which is a larger dataset.

**Extension to multi-layer transformer.** The RollingQ algorithm and the scenario we discussed are based on a single attention layer. However, as the number of layers increases in a multi-layer transformer, the situation becomes more complex than what we've analyzed so far. To effectively implement our RollingQ algorithm in such cases, we train the multi-layer attention model progressively, applying the RollingQ algorithm to a specific layer when it becomes the last attention layer in the model. For example, in a two-layer transformer fusion block, we train only the first block initially, while temporarily dropping the second block for several epochs. During this phase, we apply the RollingQ algorithm to the first block's attention layer. After a certain number of epochs, we begin training the second block as well. At this point, we apply the RollingQ algorithm to the attention layer of the second transformer block, since its query can directly influence the prediction. In a two-layer setup, the query is primarily influenced by the current input, rather than being irrelevant as in the single-layer case. Therefore, we calculate the rebalance anchor using the expectation of the query, as shown in Equation 10, and apply the rotation matrix to the query. This rotation moves the query's distribution toward the anchor region, rather than leaving it in its original position.

# C. Experiments

## C.1. Experimental Setting

**Datasets**. **CREMA-D**(Cao et al., 2014), **KineticSound**(Arandjelovic & Zisserman, 2017), and **CMU-MOSEI**(Zadeh et al., 2018) are prominent multimodal datasets for emotion recognition and sentiment analysis. CREMA-D contains 7,442 short video clips (2-3 seconds) from 91 actors, annotated with six basic emotions through crowd-sourcing. KineticSound features 31 human action classes from the Kinetics dataset, with 10-second clips manually annotated via Mechanical Turk. MOSEI, the largest of the three, includes over 23,500 sentence-utterance video clips from 1,000+ YouTube speakers, covering 250 topics with aligned language, vision, and sound features. It provides 7-class sentiment annotations and multi-label emotion intensities across six aspects.

**Backbones**. Our main experiments are conducted using 4-layer ViT(Dosovitskiy, 2020) with pretrained weights from ImageNet-21k (Ridnik et al., 2021) as the backbone, with evaluations on the KineticSound and CREMA-D datasets. For the audio and visual encoder, we adopt a ViT architecture to process multiple frames as input. Following the standard ViT approach, we extract and flatten patches from the input frames, which are then fed into the transformer layers for feature extraction. For MOSEI, we adopts vanilla transformer as encoder following settings of (Liang et al., 2021).

**Training settings**. For the KineticSound datasets, which consist of 10-second video clips, we extract frames at 1 fps and uniformly sample 3 frames per clip as visual inputs. The audio data is transformed into spectrograms of size 257×1,004 using librosa, with a window length of 512 and an overlap of 353. For CREMA-D, which contains shorter clips, we extract 1 frame per clip and process the audio into spectrograms of size 257×299, maintaining the same window length and overlap. This approach ensures consistency across datasets while leveraging the strengths of ViT for both visual and audio modalities. As for CMU-MOSEI dataset, we follow the preprocessing and settings of (Liang et al., 2021), using the extracted feature provided by (Liang et al., 2021). For training, the batch size is set to 64 for both MOSEI, CREMA-D and KineticSound. The learning rate is fixed at 1e-3, and SGD is used as the optimizer. The embedding dimensions are 120 for MOSEI and 768 for CREMA-D and KineticSound, and the cosine annealing scheduler is applied across all settings. Training is initialized from scratch for MOSEI, while pretrained models are used for CREMA-D and KineticSound, with all models trained for 30 epochs. This comprehensive setup ensures consistency and highlights the adaptability of the ViT backbone to different datasets and fusion methods.

## C.2. Validation with Attention Mask and QUAG Attention

We ablate RollingQ through mask or average attention score inspired by QUAG Attention (Rawal et al., 2023). As shown in Table 8, when masking one modality, RollingQ outperforms the baseline from 0.2 to 3%. This confirms RollingQ's ability to enhance unimodal feature quality. When using averaged attention scores, in CREMA-D (audio-dominant), vanilla MT exhibits a performance degradation of 2.3% due to over-reliance on audio features. For Kinetic-Sound (relative balance), vanilla MT's performance increases by 1.2% since the model learns an unreasonable attention score due to the self-reinforcing cycle. Conversely, RollingQ maintains stable performance (with less than 1% drop), demonstrating it's leveraging complementary modality information and robustness.

*Table 8.* Ablation on Kinetic-Sound dataset. Mask V refers to force the attention score of visual modality to 0 by adding attention mask, and vice versa. Besides, "uni_A" and "uni_V" refers to the performance of training solely with unimodal model.

| Method | Vanilla MT | Vanilla MT+RollingQ | uni_A | uni_V |
|---|---|---|---|---|
| A (Mask V) | 46.6 | 47.4 (↑ 0.8) | 53.9 | - |
| V (Mask A) | 40.3 | 42.1 (↑ 1.8) | - | 57.0 |
| origin | 67.0 | 69.3 | - | - |

Furthermore, to explore how RollingQ influence the intra- and inter-modality interactions learned by model, we explore on the baseline with transformer fusion blocks and conduct complete QUAG (Rawal et al., 2023) tests (unimodal, crossmodal, audio-avg, and video-avg) with and without RollingQ. As shown in Table 9, the RollingQ algorithm exhibits more performance drops around 2.7% to 5.0% across all types of QUAG tests compared to baseline. This indicates that RollingQ can not only fully leverage both modalities faithfully but also learn comprehensive multimodal interactions.

*Table 10.* Results on OOD benchmarks with HMDB51 as In-Distribution dataset following the preprocessing and settings of Multi-OOD(Dong et al., 2024). General Entropy (GEN) is used to obtain the confidence for OOD data.

| OOD-Dataset | HMDB | | | UCF | | |
|---|---|---|---|---|---|---|
| Metrics/Methods | IDAcc ↑ | FRR95 ↓ | AUROC ↑ | IDAcc ↑ | FRR95 ↓ | AUROC ↑ |
| Vanilla MT | 51.6 | 77.3 | 64.4 | 48.7 | 77.9 | 70.4 |
| Vanilla MT + RollingQ | 54.0 (↑ 2.4) | 74.9 (↓ 2.4) | 68.6 (↑ 4.2) | 51.8 (↑ 3.1) | 76.9 (↓ 1.0) | 73.2 (↑ 2.8) |

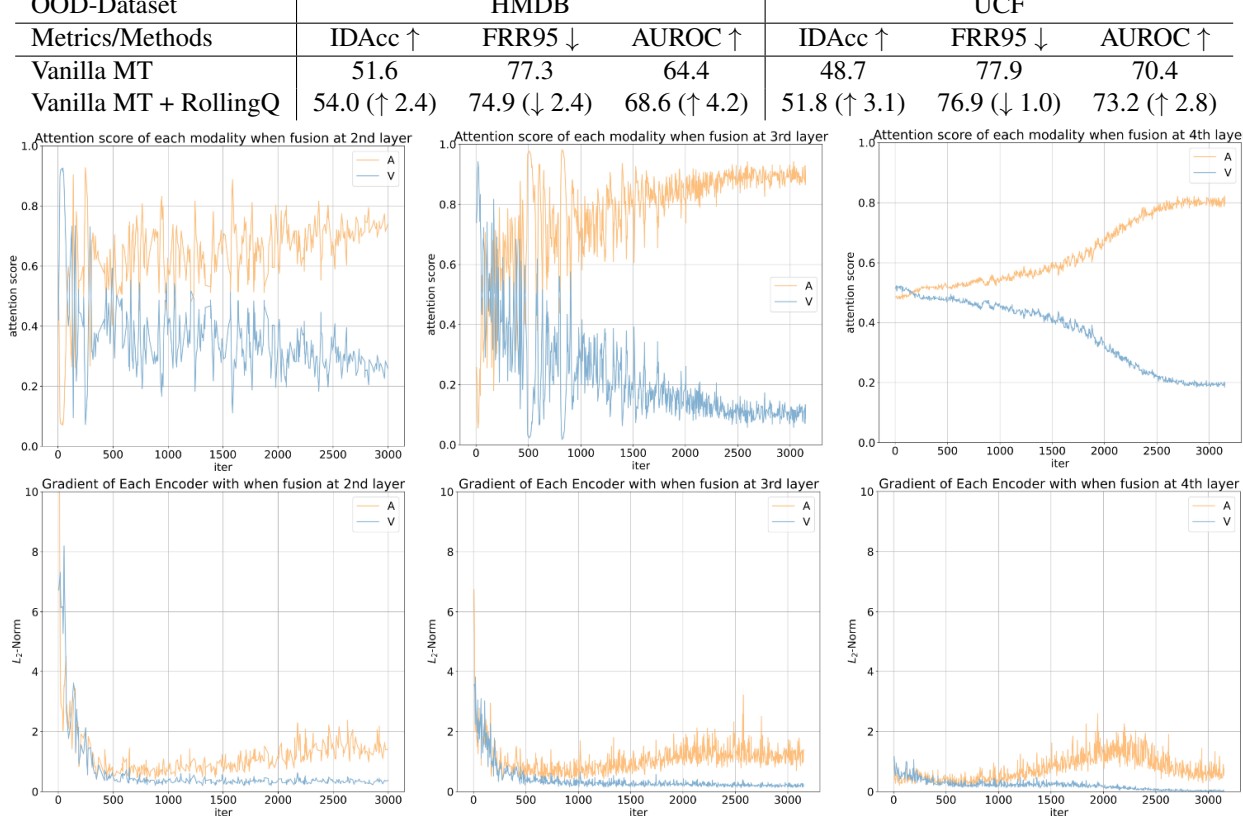

*Figure 8.* The $L_2$-Norm of gradient of the audio encoder and the visual encoder and attention score across modalities on CREMA-D and Kinetic-Sound datasets.

*Table 9.* QUAG operations (Rawal et al., 2023) on Kinetic-Sound dataset, the more drops indicates the more leverage. uni_A and uni_V refers to the performance of training solely with unimodal model.

| Method | baseline | baseline+RollingQ | uni_A | uni_V |
|---|---|---|---|---|
| origin | 69.1 | 70.1 | - | - |
| unimodal | 67.8 (↓ 1.3) | 63.8 (↓ 6.3) | - | - |
| crossmodal | 65.5 (↓ 3.6) | 63.8 (↓ 6.3) | - | - |
| avg-audio | 64.8 (↓ 4.3) | 62.9 (↓ 7.2) | 53.9 | - |
| avg-visual | 65.6 (↓ 3.5) | 62.1 (↓ 8.0) | - | 57.0 |

### C.3. Verification on OOD Benchmarks

OOD is challenging task where test distribution exhibits distribution shifts compared to training distribution. It is a effective way to further verify the dynamic property of multimodal transformer to adaptively assign attention score. Hence, we conducts multimodal OOD experiments following (Dong et al., 2024). Both Far-OOD and Near-OOD cases discussed in (Dong et al., 2024) are involved here to present a comprehensive evaluation. Specifically, we use HMDB51 for Near-OOD test and use HMDB51 and UCF-101 for Far-OOD test.

As shown in Table 10, our RollingQ algorithm outperforms the baseline on all metrics, showing that it can ease the modality bias with strong generailzation ability and effectively fuse the multimodal feature even for out-of-distribution samples.

### C.4. Extension to More Fusion Paradigms

Multimodal transformers are designed under different fusion paradigms such as early fusion, mid fusion and late fusion. We perform further validation of the effectiveness of our proposed RollingQ algorithm under earlier fusion paradigms.

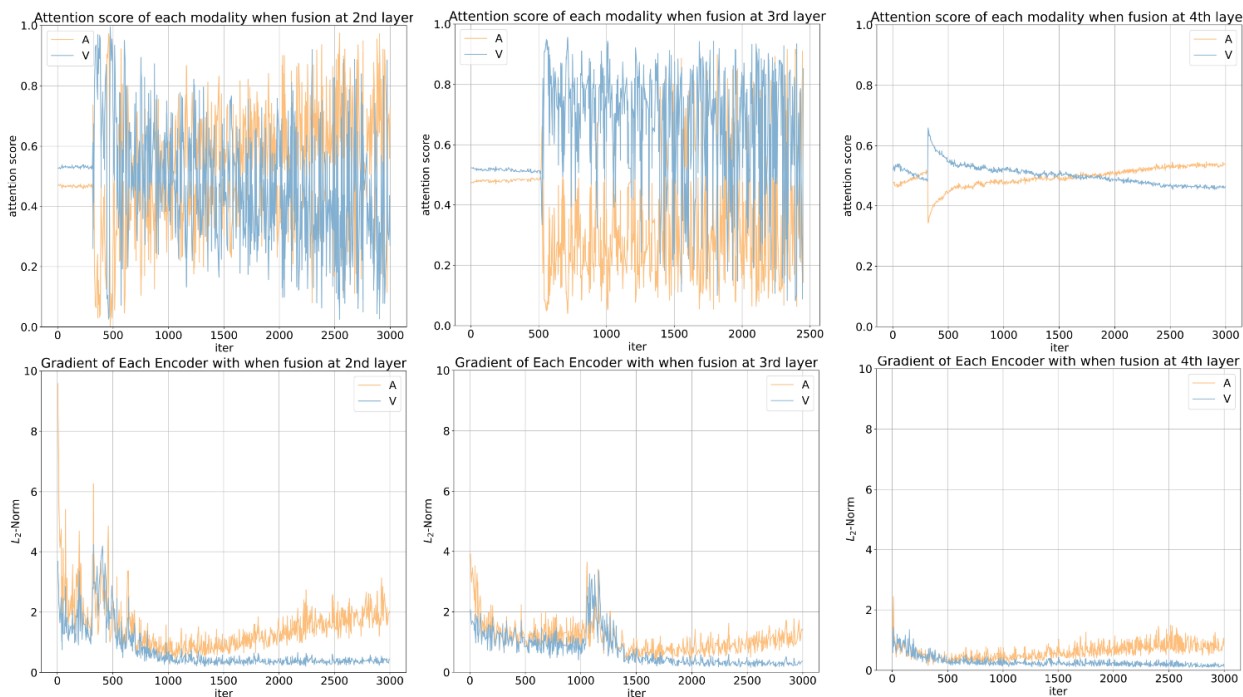

*Figure 9.* The $L_2$-Norm of gradient of the audio encoder and the visual encoder and attention score across modalities on CREMA-D and Kinetic-Sound datasets when implement RollingQ.

*Table 11.* Performance of vanilla multimodal transformer (Vanilla MT) and RollingQ across different fusion layers, where "2nd" refers to fusion from the 2nd transformer block.

| Fusion Layer | Vanilla MT | Vanilla MT + RollingQ |
|:---:|:---:|:---:|
| 2nd | 42.3 | 42.5 (↑ 0.2) |
| 3rd | 41.1 | 42.2 (↑ 1.1) |
| 4th | 41.4 | 43.7 (↑ 2.3) |

Specifically, we adopts the same 4-layer ViT backbone on CREMA-D dataset but start to fuse features from two modalities by different layer from 2nd, 3rd and 4th layer. Firstly, we monitor the gradient of unimodal encoders and the attention which are the two driving factors of the self-reinforcing cycle. As shown in Figure 8, the gradient difference declines and the attention becomes balanced and more unstable as the fusion becomes earlier. Meanwhile, we monitor the gradient and attention after applying RollingQ. As shown in Figure 9, after applying RollingQ, both the attention score and the gradients across modalities are closer, indicating the ability of RollingQ to rebalance the feature quality and revive the cooperation dynamics provided by attention mechanism.

Afterwards, we conduct systematically comparison under different fusion layers. As shown in Table 11, The results reveals that our method can be applied to earlier fusion paradigms achieving better performance around (2.3% / 1.1% / 0.2%), but it's more suitable for late fusion paradigm as we discussed the imbalance phenomenon is not significant under earlier fusion paradigms.

