# OpenReview forum: "RollingQ: Reviving the Cooperation Dynamics in Multimodal Transformer"
_ICML.cc/2025/Conference — ICML 2025 poster_

### Official Review · Reviewer_gn7r · 2025-03-11

**Overall Recommendation:** 4

**Summary:**

The paper analyzes an important problem of modality biases in multimodal learning setup. The authors find that the dynamic property of attention is lost during multimodal training; that is, rather than weighing the modalities per-instance, the models just focus on a single (biased) modality, which is overemphasized during training, leading to distribution gaps. To this, author propose Query Rebalanced Rotation (QRR) algorithm that rebalances the query to “revive” the dynamic property of attention by rotating the query vector towards an anchor that provides higher weights to the unbiased modality, thus reducing the modality bias in attention.

**Claims And Evidence:**

I am not convinced why the rotation of the query vector towards the defined anchor would help reduce the modality bias. Further, the results supporting this claim are not consistent and thorough (elaborated below)

**Essential References Not Discussed:**

- The related works section of the paper does not discuss the necessary works on unimodal and multimodal biases in datasets like [1], that is extremely relevant for such studies
- Other important works like QUAG and QUAG-attention [2] perform very similar analysis on average key per modality and should be (a) acknowledged in the relevant works, and (b) utilized as a verification that QRR is indeed leveraging both the modalities. Further, datasets like CLAVI [2] and Perception Test [3] could be used as debiased test datasets


[1] Buch, Shyamal, et al. "Revisiting the "video" in video-language understanding." CVPR 2022. \
[2] Rawal, Ishaan Singh et al., “Dissecting Multimodality in VideoQA Transformer Models by Impairing Modality Fusion”. ICML 2024 \
[3] Pătrăucean, Viorica et al., “Perception Test: A Diagnostic Benchmark for Multimodal Video Models”. NeurIPS 2023

**Experimental Designs Or Analyses:**

For CREMA-D dataset, the authors were able to achieve high performance with just a single frame and the audio. This is concerning. This all the more emphasizes that the benchmarks have biases. Being truly multimodal means that the modal *needs* to leverage both the modalities faithfully. However, in this case it could be leveraging weak biases in the multiple modalities, and combining them to produce the answer. Therefore, I strongly suspect that while QRR might re-weigh the attention towards “unbiased modality”, the small gains in performance in some cases might be because t inadvertently makes the model leverage multimodal biases

**Methods And Evaluation Criteria:**

- It is known that the quality of multimodal representations is heavily influenced by the fusion strategy (early/mid/late). In the paper, QRR is based on a very simple late fusion strategy. However, many works adopt early fusion strategies. The paper misses out on (a) systematic comparison with these methods, (b) evolution of cooperative dynamics in these setups, and (c) difference in QRR behaviours in these setups.
- Does QRR work for the right reasons? The paper, while claims multimodal debiasing, does not provide any results on challenging OOD benchmarks. Multimodal datasets can have biases, and there are multiple challenging benchmarks for evaluating the true multimodal performance of the models. Therefore, performance on biased benchmarks is not representative of the unbiasedness property of the model. The closest to this result in the paper are the noise_T experiments, which are more of a sanity check than an OOD benchmark.

**Other Comments Or Suggestions:**

- Should it be q_b instead of q_r in figure 2?
- For most of the graphs, the legend is not explained. Would be nice to have detailed captions.
- The running header of the paper isn’t updated

**Other Strengths And Weaknesses:**

**Strengths**
1. The finding that static fusion techniques might be on-par with dynamic fusion like attention based techniques is indeed surprising and re-affirms the difficulty in taming multimodal self-attention
2. The paper is aiming to tackle an extremely important problem of biased representation is multimodal learning which can be proven very valuable to the community
3. QRR is a generic method that can be applied to multiple domains
3. The paper is cohesively written and easy to follow

**Weaknesses** \
Lack of convincing results:
1. No results on OOD benchmarks (see above) and validation tests (see above)
2. No consistent and significant improvement in accuracy

**Questions For Authors:**

- What is the effect of batch size on the QRR? How sensitive is it?
- What is the overhead of QRR (computational complexity and/or increase in run-time/FLOPs)?

**Relation To Broader Scientific Literature:**

I believe that the findings of the paper regarding unimodal biases are in congruence with the related works in the domain

**Theoretical Claims:**

Not applicable

---

> ### Author Rebuttal · Authors · 2025-04-01
>
> Dear reviewer gn7r,
>
> **Thanks a lot for your valuable review, suggestions, and questions.**
>
> **Q1: Extension to more fusion paradigms**
>
> > Q1.a: Analysis of cooperation dynamics.
>
> To analyze cooperation dynamics across fusion methods, we monitor the gradient of unimodal encoders and the attention which are the two driving factors of the self-reinforcing cycle. As shown in [Figure 2](https://anonymous.4open.science/r/ICML-2025-Rebuttal-07BE/Figure2.png), the gradient difference declines and the attention becomes balanced and more unstable as the fusion becomes earlier.
>
> > Q1.b: Systematic comparison and QRR behaviors.
>
> Following descriptions in Appendix B, we train the model progressively and apply QRR. The comparison results are shown in [Table 1](https://anonymous.4open.science/r/ICML-2025-Rebuttal-07BE/Table1.png), where Vanilla MT denotes the vanilla multimodal transformer. The results reveals that our method can be applied to earlier fusion paradigms achieving better performance around (2.3% / 1.1% / 0.2%), but it's more suitable for late fusion paradigm.
>
> **Q2: Verification and validation.**
>
> > Q2.a: Results on OOD benchmarks.
>
> Thanks for providing a novel aspect for testing QRR. We adopt MultiOOD [1]. As shown in [Table 2](https://anonymous.4open.science/r/ICML-2025-Rebuttal-07BE/Table2.png), our QRR algorithm outperforms the baseline on all metrics, showing that our QRR algorithm can ease the modality bias during training dynamics.
>
> [1] Hao, D., et al. "MultiOOD: Scaling Out-of-Distribution Detection for Multiple Modalities", NeurIPS 2024.
>
> > Q2.b: Comparison with QUAG[2]
>
> QUAG is designed based on VideoQA tasks using **frozen unimodal encoders**, which sets it apart from our work. Besides, QUAG performs analysis on averaging attention score but our analysis focus on feature space of key, query in attention.
>
> [2] Rawal, Ishaan Singh, et al., “Dissecting Multimodality in VideoQA Transformer Models by Impairing Modality Fusion”. ICML 2024
>
> > Q2.b: QRR might not leverage both modalities due to modality bias.
>
> In this work, we focus on modality bias that one modality dominates the training process[3], leading to inequality modality feature, unreasonable attention and sub-optimal performance. Hence, we propose QRR to balance the learning of unimodal encoders, revive the cooperation dynamics to leverage both modality.
>
> We ablate QRR through mask or average attention score inspired by QUAG[2]. As shown in [Table 3,4](https://anonymous.4open.science/r/ICML-2025-Rebuttal-07BE/Table3%20&%204.png), when masking one modality, QRR outperforms the baseline from 0.2 ~ 3%. This confirms QRR's ability to enhance unimodal feature quality. When using averaged attention scores, in CREMA-D (audio-dominant), vanilla MT exhibits a performance degradation of 2.3% due to over-reliance on audio features. For Kinetic-Sound (relative balance), vanilla MT's performance increases by 1.2% since the model learns an unreasonable attention score due to the self-reinforcing cycle. Conversely, QRR maintains stable performance (with <1% drop), demonstrating it's leveraging complementary modality information and robustness.
>
> [3] Peng, X., et al. "Balanced multimodal learning via on-the-fly gradient modulation." CVPR 2022.
>
> **Q3: Supporting details on the stability, efficiency, and performance**
>
> > Q3.a: Batch size ablation.
>
> Through batch size ablation on CREMA-D in [Table 5](https://anonymous.4open.science/r/ICML-2025-Rebuttal-07BE/Table5.png), QRR maintains stable performance improvements (0.7%-3.1%) compared to baseline from 16 to 256 batch size, demonstrating remarkable training stability.
>
> > Q3.b: Limited performance improvements.
>
> Compared to methods requiring additional parameters and specialized modules, QRR only requires：
> - Parameter increase: **1.0%**
> - FLOPs increase: **0.1%**
>
> as shown in [Table 7](https://anonymous.4open.science/r/ICML-2025-Rebuttal-07BE/Table7.png), while accuracy achieve **3.1%** improvements on CREMA-D and **2.3%** improvements on Kinetic-Sound compared to vanilla multimodal transformer. , QRR is a simple yet effective method that achieves comparable and even better results with less computation cost.
>
> **Q4: Details on paper writing and figures**
>
> > Q4.a: Notation in figure 2.
>
> In a single fusion layer setting, $q_b$ and $q_r$ are the same. As the number of fusion layers increases, the query $q$ becomes influenced by the input and is no longer a static value. Hence, for each input $q$, the rotation matrix $q_b$ can not ensure to rotate the $q$ to $q_b$, but should be located near this region.
>
> > Q4.b: Detailed captions, missing of related works[5,6] and header of the paper.
>
> Thanks for pointing this out, we'll add relative discussion in our revised manuscript.
>
> [5] Buch, Shyamal, et al. "Revisiting the "video" in video-language understanding." CVPR 2022.
>
> [6] Pătrăucean, Viorica et al., “Perception Test: A Diagnostic Benchmark for Multimodal Video Models”. NeurIPS 2023

---

> > ### Comment · Reviewer_gn7r · 2025-04-04
> >
> > Dear authors, thanks for the detailed rebuttal.
> >
> > **Q1**: Thanks for the additional experiment. The results from Figure 2 are quite interesting. However, it is incomplete. For a complete picture, it would be nice to have the change in graphs after applying QRR. Table 1 results also seem interesting. Probably it is because previous works have found increased modality bias in the late fusion strategy, therefore the "healing" effect of QRR is most drastic there.
> >
> > **Q2 a.** I must admit the the results are not very convincing (the gap is too low) however the trend is in general consistent with the authors' intuitions.
> >
> > **Q2 b.** The masking results are interesting that both the modalities are being leveraged more individually. However, multimodality is beyond individual modality but also consider synergistic and redundant information arising through modality interactions. Therefore I suggested QUAG. However, the QUAG results are incomplete (all unimodal, crossmodal, audio-avg and video-avg cases). It'd be nice to have it for completeness and corroborate the results.
> >
> > **Q3a.**: The results are interesting. Did you investigate why the effect of QRR is minimal for higher batch sizes? It'd be nice to investigate it further, or atleast mention it in the future works.
> >
> >
> > The additional experiments have increased my confidence it authors' works. Even though the performance improvement is not a lot, I got many insights from their analysis. However, I do agree there are some loose ends in the paper that could be polished more.
> >
> >  To this, I increase my score to 3. I hope the authors can follow-up on the questions and add the new experiments to their paper.
> >
> >
> > ###EDIT####
> >
> >
> > After reviewing the authors' latest rebuttal I am more confident of authors' works. I appreciate their prompt response and detailed experiments of their method validated on OOD benchmarks, learning dynamics and QUAG. While the increment in performance is not drastic, I think the work is insightful enough to garner interest of the multimodal community. To this, I am increasing my score to 4. I hope the authors can add these experiments to their paper.
> >
> > Thanks.
> >
> >
> > ###########
> >
> >
> > Regards.

---

> > > ### Author Response · Authors · 2025-04-06
> > >
> > > Dear gn7r,
> > >
> > > **Thanks for your thoughtful comment and affirmation of our work. Your constructive suggestions have been invaluable in helping us refine and polish our work.** Hence, we have carefully followed up on the questions raised and expanded our experiments, hoping to address your concerns from a more comprehensive perspective.
> > >
> > > > Q1: (**Extension to more fusion paradigms**) need complete picture with changes after applying QRR.
> > >
> > > Thanks for pointing this out. To address this, we added visualization with gradients and attention scores. As shown in [Figure2-addition](https://anonymous.4open.science/r/ICML-2025-Rebuttal-07BE/Figure2-addition.png), after applying QRR, **both the attention score and the gradients across modalities are closer**, indicating the ability of QRR to ease the modality bias. Besides, some previous works indeed reveal that the modality bias is increased in the late fusion paradigm [1].
> > >
> > > [1] Yedi, Z., et al. "Understanding Unimodal Bias in Multimodal Deep Linear Networks.", ICML 2024.
> > >
> > > > Q2.a:  (**OOD benchmarks**) OOD Results.
> > >
> > > Thanks for mentioning this. For multimodal OOD detection, improving the representation quality [2] and designing better score functions for OOD prediction are two strategies and closely related [3,4]. In this work, QRR mainly focuses on providing high-quality multimodal representation. In our initial reply, to obtain the results at a faster rate, we use the simplest Maximum Softmax Prediction (MSP) as the score function, but which is an underestimation of true representation quality [4]. QRR still brings improvements in this situation. At this time, we conduct experiments on more effective score functions, Energy [3] and GEN [4], which could more accurately reflect the representation quality. As shown in [Table2-update](https://anonymous.4open.science/r/ICML-2025-Rebuttal-07BE/Table2-update.png), comparing to baseline Vanilla MT, QRR achieves around **2.4%-3.1%** increase on ID-Acc, **0.9%-2.4%** decrease on FRR95, and **2.8%-4.2%** increase on AUROC. These suggest that QRR can enhance multimodal representation and leverage information from both modalities.
> > >
> > > We sincerely thank you for providing such a valuable perspective on testing the QRR under more comprehensive and challenging circumstances, which further demonstrates our effectiveness. We'll add these results to the revised manuscripts.
> > >
> > > [2] Hao, D., et al. "MultiOOD: Scaling Out-of-Distribution Detection for Multiple Modalities", NeurIPS 2024.
> > >
> > > [3] Liu, W., et al. "Energy-based out-of-distribution detection.", NeurIPS 2020.
> > >
> > > [4] Liu, X., et al. "Gen: Pushing the limits of softmax-based out-of-distribution
> > > detection.", CVPR 2023.
> > >
> > > > Q2.b: (**QUAG test**) QUAG results are incomplete (all unimodal, crossmodal, audio-avg, and video-avg cases).
> > >
> > > Thanks for providing a valuable validating method to evaluate our model from a multimodal interaction perspective. To explore intra- and inter-modality interactions, we explore on the baseline with transformer fusion blocks and conduct complete QUAG tests (unimodal, crossmodal, audio-avg, and video-avg) with and without QRR. As shown in [Table8](https://anonymous.4open.science/r/ICML-2025-Rebuttal-07BE/Table8.png), the QRR algorithm exhibits **more performance drops around 2.7%-5.0% across all types of QUAG tests** compared to baseline. This indicates that QRR can not only fully leverage both modalities faithfully but also learn comprehensive multimodal interactions.
> > >
> > > > Q3.a: (**Batch size ablation.**) why the effect of QRR is minimal for higher batch sizes?
> > >
> > > Thanks for your so nice and responsible reviews which helps us to analyze the experiments. Since QRR is a sample-wise modulation, we consider that it might be caused by the reduction in sample-wise variation when batch sizes grow, inadvertently harming the fitness between the rotation matrix and data on a per-sample basis. Thanks for pointing out this, we'll mention this in future work to promote further discussion in our revised manuscripts.
> > >
> > > Once Again, **we greatly appreciate your affirmation of our analysis**, we also believe that QRR has the potential to revive the cooperation dynamics of transformers and be meaningful and valuable to the community.
> > >
> > > If you have any further concerns or suggestions, please feel free to share them, and we will carefully consider and revise our work accordingly. Thanks a lot for your contribution, we'll add these experiments and analyses to our revised manuscripts.
> > >
> > > \#\#\#\# EDIT \#\#\#\#
> > >
> > > **We sincerely appreciate your confidence in our work and your affirmation of our contribution to the multimodal community.** Thanks greatly for your invaluable and constructive suggestions helping us to polish our work. We'll add these experiments to our paper.
> > >
> > > Best regards.
> > >
> > >  \#\#\#\#\#\#\#\#\#\#\#\#

---

### Official Review · Reviewer_FKSM · 2025-03-14

**Overall Recommendation:** 3

**Summary:**

This paper focuses on fusion strategies in multimodal transformers, identifies issues in dynamic fusion, proposes the QRR algorithm, and validates its effectiveness in restoring cooperation dynamics and improving performance through experiments

**Claims And Evidence:**

The paper's claims are supported by some evidence, but the universality in broader scenarios lacks sufficient proof as experiments are based on specific datasets and settings

**Essential References Not Discussed:**

none

**Experimental Designs Or Analyses:**

The experimental design is comprehensive, yet the analysis lacks indepth statistical methods to determine result significance and stability.

**Methods And Evaluation Criteria:**

The QRR algorithm is reasonably designed, and the benchmark datasets are suitable. However, the evaluation lacks indepth analysis of dataset characteristics.

**Other Comments Or Suggestions:**

none

**Other Strengths And Weaknesses:**

**Strengths:** The research addresses a problem of practical significance by proposing a solution to the fusion challenges encountered in the practical application of multimodal Transformers. The QRR algorithm is straightforward yet effective, requiring no additional training loss and enhancing performance without increasing model complexity, which demonstrates its innovativeness.

**Weaknesses:** I feel that the paper lacks in-depth analysis and does not adequately validate its assumptions. The experimental section employs too few benchmarks, and I would like to see whether the proposed method can provide assistance to existing multimodal models.

**Questions For Authors:**

none

**Relation To Broader Scientific Literature:**

The QRR algorithm offers a novel approach to addressing the issue of modality imbalance, differing from prior methods that optimized single-modal encoders, thereby extending the existing research framework and providing new insights for multimodal fusion.

**Theoretical Claims:**

The theoretical analysis is clear.

---

> ### Author Rebuttal · Authors · 2025-04-01
>
> Dear reviewer FKSM,
>
> **We appreciate your time and great efforts in reviewing.**
>
> We carefully considered your comments on the validation of assumptions, lack of in-depth analysis and extension to benchmarks and methods and conducted corresponding experiments and theoretic analysis.
>
> **Q1: The validation of the deactivation of the cooperation dynamics assumption is not adequate since the datasets and settings are limited.**
>
> Thanks for pointing out. In the previous version, we provide visualization and analysis of Kinetic-Sound for this assumption. To holistically and systematically validate our assumption, which mainly states that the modality bias in the multimodal training process triggers a self-reinforcing cycle that leads to inequality in feature quality and unreasonable attention score, we conduct experiments on more datasets including CREMA-D, Kinetic-Sound, CMU-MOSEI(A+T), CMU-MOSEI(V+T), UCF-101, HMDB51, whose modality ranging from audio, RGB, text and optical flows and exhibiting variation in modality sequence length. As shown in [Figure 1](https://anonymous.4open.science/r/ICML-2025-Rebuttal-07BE/Figure1.png), with different modalities combinations and dataset scales, the unreasonable attention score attribution agree with our previous study. Besides, the visualization of average key distribution, where the noise input consistently has higher cosine similarity, further validates our assumption and theoretic analysis towards cooperation dynamics under imbalance multimodal learning.
>
> **Q2: Lacks in-depth analysis**
>
> > Q2.a: in-depth analysis of self-reinforcing cycle.
>
> In addition to the observation of attention score during training dynamics, we monitor the gradient of unimodal encoders to verify our proposition of a self-reinforcing cycle. As shown in [Figure 3](https://anonymous.4open.science/r/ICML-2025-Rebuttal-07BE/Figure3.png), the gradient of both modalities's encoder are similar during the start time. However, the biased modality (audio) encoder increases significantly during the mid-stage, when the attention score begins to accumulate in the biased modality. Later, both of the gradients drop due to the minimization of total loss. This observation further verifies our theoretical analysis and ensures the existence of the self-reinforcing cycle.
>
> Moreover, from the perspective of fusion settings, we visualize the gradient and distribution of attention scores on different modalities under comprehensive fusion paradigms. As shown in [Figure 2](https://anonymous.4open.science/r/ICML-2025-Rebuttal-07BE/Figure2.png), even as the starts of the fusion layers varied, the gradients of the biased unimodal encoder and attention score towards biased modality consistently exhibits considerable greater than the unbiased one, verifying our assumption and analysis of the self-reinforcing cycle.
>
> > Q2.b: in-depth analysis of significance and stability.
>
> To evaluate the stability and result significance of our method. We conduct repetitive experiments over CREMA-D and Kinetic-Sound with only differences in random seeds. By applying Pearson correlation analysis as shown in [Table 6](https://anonymous.4open.science/r/ICML-2025-Rebuttal-07BE/Table6.png), with 0 denotes vanilla multimodal transformer while 1 denotes applying QRR algorithm, the coefficient is 0.765 for CREMA-D and 0.698 for Kinetic-Sound indicates its statistically significant on performance improvements with p-value < 0.01 ensures the confidence of the analysis.
>
> Besides, ablation on batch size ensures the stability of our method. As results shown in [Table 5](https://anonymous.4open.science/r/ICML-2025-Rebuttal-07BE/Table5.png), with batch size varying from 16 to 256 on CREMAD, the QRR consistently outperforms baseline methods by **0.7% ~ 3%**, revealing the stability of our method.
>
> **Q3: Extension**
>
> > Q3.a: Extension to more benchmarks
>
> In addition to existing benchmarks and testing, We adopt MultiOOD [1] for the Out-of-Distribution benchmark and further validate the performance of our QRR method. Here, we consider both near-OOD and far-OOD scenarios discussed in MultiOOD. As results shown in [Table 2](https://anonymous.4open.science/r/ICML-2025-Rebuttal-07BE/Table2.png), our QRR algorithm consistently outperforms the baseline on all metrics and fully tests the effectiveness of our method from another perspective.
>
> [1] Hao, D., et al. "MultiOOD: Scaling Out-of-Distribution Detection for Multiple Modalities", NeurIPS 2024.
>
> > Q3.b: Combine QRR with existing methods
>
> Due to the time constraints, we haven't done enough attempts to combine our method with existing multimodal methods, which would be conducted during discussion period.
>
> If you have any valuable questions or constructive suggestions that could help you understand our work, please tell us and help to produce a higher quality paper.

---

> > ### Comment · Reviewer_FKSM · 2025-04-08
> >
> > The responses has addressed my concerns, and I’m willing to raise my score.

---

> > > ### Author Response · Authors · 2025-04-08
> > >
> > > **We sincerely appreciate your affirmation of the innovativeness and significance of our work**, and we're more than happy to know that your concerns have been addressed. Thanks for your valuable time reviewing and invaluable suggestions to improve our work.

---

### Official Review · Reviewer_61wP · 2025-03-14

**Overall Recommendation:** 3

**Summary:**

The paper identifies the issue of the self-reinforcing cycle toward the majority modality in multimodal learning. To address this, the authors propose a query rebalance rotation method that disrupts the cycle and rebalances the attention mechanism. Experimental results and visualizations demonstrate the effectiveness of the proposed method.

**Claims And Evidence:**

Overall, the claims are well-supported. For example, in the introduction, each claim is supported by recent references or empirical results. Additionally, the investigation into the superior performance of static fusion over dynamic fusion is interesting and highlights the motivations of the paper.

**Essential References Not Discussed:**

-

**Experimental Designs Or Analyses:**

1. In table 1, compared with baselines, the improvements are limited. On CREMA-D and MOSEI, the proposed method achieves only a 0.2% gain, while on Kinetic-Sound, its performance is even worse than the baselines.
2. Given the limited improvement, significance tests are necessary to determine whether the gain is statistically meaningful.
3. Section 4.3 is interesting, and Figure 4 is clear and effectively demonstrates the impact of the proposed QRR module in enhancing multimodal learning.

**Methods And Evaluation Criteria:**

-

**Other Comments Or Suggestions:**

Overall, I am inclined to accept the paper, as it presents insightful findings and visualizations, even though the proposed method does not outperform the state-of-the-art.

**Other Strengths And Weaknesses:**

-

**Questions For Authors:**

Please refer to my comments above.

**Relation To Broader Scientific Literature:**

-

**Theoretical Claims:**

1. The authors mention that as training progresses, the superior modality gains more attention and is better optimized. However, is this always the case, even in the late training stage when training is nearly converged? I believe that even in the early stages, the superior modality, having a higher gradient for backpropagation, would converge more quickly and reach a (sub-)optimal point, at which its gradient becomes smaller than that of the weaker modality. Could you clarify this reasoning?
2. The setting in Section 3.2 appears to be under cold-start training, meaning the models are trained from scratch. As far as I know, CREMA-D and CMU-MOSEI are highly biased multimodal datasets, where the feature quality of one modality is significantly better than the others. What if multimodal transformers were pre-trained on more balanced and diverse datasets? Would the self-reinforcing cycle still persist in that case?

---

> ### Author Rebuttal · Authors · 2025-04-01
>
> Dear reviewer 61wP,
>
> **Thank you very much for your affirmation and constructive comments.**
>
> We carefully considered your comments and conducted corresponding experiments.
>
> **Q1: The change of gradients during training.**
>
> Thank you for your question. From the optimization perspective, previous works on imbalanced multimodal learning have extensively discussed the evolution of gradients under a late fusion structure with concatenation or summation fusion, which shows similarities to our setup[1, 2]. A well-established observation is that the superior modality always gains more optimization momentum, as its gradient is higher than that of the weaker modality.
>
> However, our setup differs from previous simpler structures where we adopts more complex transformer blocks for multimodal fusion, and your question prompted us to create a clearer visualization to provide more convincing evidence for our theoretical analysis.
>
> As shown in [Figure 3](https://anonymous.4open.science/r/ICML-2025-Rebuttal-07BE/Figure3.png), the gradient of the audio encoder increases significantly during the mid-stage, when the attention score begins to accumulate in the biased modality. This results in a noticeable gap between the modalities. As the total loss decreases over time, the gradients of both modalities drop, but their relative relationship remains. This further validates our theoretical analysis, showing that the biased modality consistently receives more optimization momentum. The gradient of the weaker modality never exceeds that of the superior modality, which can be explained by Equations 7 and 8 in our paper. Since the only difference in gradients for each modality is $\frac{\partial h_i}{\partial z_i^m}$ and the loss is the multimodal loss, where "one modality becoming converged" equals "the multimodal model becoming converged," the total loss becomes very small. This small loss cannot provide enough momentum for the superior modality to optimize effectively. Hence, the gradient of the superior modality will be consistently greater than the weak one.
>
> [1] Peng, X., et al. "Balanced multimodal learning via on-the-fly gradient modulation." CVPR 2022.
>
> [2] Fan, Y., et al. "Pmr: Prototypical modal rebalance for multimodal learning." CVPR 2023
>
> **Q2: Cold-Start and pretraining's influence on self-reinforcing cycle.**
>
> Thank you for your question. As discussed in the Experiments section, we are not using cold-start but using a 4-layer ViT-B/16 as the backbone, initialized with pre-trained weights from ImageNet-21k. For the MOSEI dataset, we use the Vanilla Transformer without pretraining. The results shown in Table 1 and the visualizations in Figure 4 demonstrate that even under pretraining conditions, the self-reinforcement cycle still exists and harms the overall performance of multimodal transformers.
>
> **Q3: Limited Improvements**
>
> > Q3.a: Significance tests to prove the gain is statistically meaningful.
>
> Thank you for your suggestion. To prove the significance and stability of QRR, we selected 10 random seeds and conducted repeated experiments with the same settings on the CREMA-D and Kinetic-Sound datasets. By applying Pearson correlation analysis as shown in [Table 6](https://anonymous.4open.science/r/ICML-2025-Rebuttal-07BE/Table6.png), with 0 denotes vanilla multimodal transformer while 1 denotes applying QRR algorithm, the coefficient is 0.765 for CREMA-D and 0.698 for Kinetic-Sound indicates its statistically significant on performance improvements with p-value < 0.01 ensures the confidence of the analysis. We'll further accomplish a more systematic analysis of pretraining's influence during the discussion period.
>
> > Q3.b: Limited improvements.
>
> Thanks for mentioning this. However, the QRR, which requires only:
> -  **1%** increase on parameters
> -  **0.1%** on GFLOPs
>
> As shown in [Table 7](https://anonymous.4open.science/r/ICML-2025-Rebuttal-07BE/Table7.png). It achieves considerable improvements on its baseline vanilla multimodal transformer (accuracy increase by **3.1%** on CREMA-D and by **2.3%** on Kinetic-Sound), while gaining comparable and even better performance over other specially designed transformer architecture with much more requirements on computation resources and increased complexity. Hence, the improvements are significant especially taking the computation and time cost into consideration.
>
> As for CMU-MOSEI, we acknowledge this is an difficult dataset for multimodal sentiment analysis. With all of our comparison methods achieving limited improvements and previous research [3] with three modalities input could only achieve around 1.5% improvements, our QRR algorithm performs relatively significant increment.
>
> [3] Paul Pu Liang et al. "Multibench: Multiscale Benchmarks for Multimodal Representation Learning." NeurIPS, 2021
>
> If you have any valuable questions or constructive suggestions that could help you understand our work, please tell us and help to produce a higher quality paper.

---

> > ### Comment · Reviewer_61wP · 2025-04-02
> >
> > Thanks authors for the rebuttal. Most of my concerns are well-addressed especially the performance part. I will maintain my ratings of weak accept for it still represents my overall impression of the paper's contribution and significance at this time.

---

> > > ### Author Response · Authors · 2025-04-03
> > >
> > > We sincerely appreciate your constructive suggestions and insightful feedback, which have been invaluable in helping us refine and improve our work. Additionally, **we are grateful for your affirmation and positive comments regarding the significance and contribution of our work.**

---

### Decision · Program_Chairs · 2025-05-01

**Decision:**

Accept (poster)

**Comment:**

This paper identifies a weakness in attention-based multimodal fusion, that is, when one modality becomes biased, it disrupts the dynamic weighting in Transformers. The authors propose Query Rebalance Rotation to rebalance attention queries to prevent over-reliance on any single modality. QRR restores attention flexibility and improves multimodal learning performance, especially when modality quality varies. Experimental results show its superior performance when compared to other SOTA works. Based on the comments, recommendations, and post-rebuttal discussion, the authors’ point-to-point rebuttal addressed most of these concerns so that no significant issues remain. This paper is acceptable in its current form.